# Bearing Fault Diagnosis Method Based on Convolutional Neural Network and Knowledge Graph

**DOI:** 10.3390/e24111589

**Published:** 2022-11-02

**Authors:** Zhibo Li, Yuanyuan Li, Qichun Sun, Bowei Qi

**Affiliations:** School of Electronic and Electrical Engineering, Shanghai University of Engineering Science, Shanghai 201620, China

**Keywords:** fault diagnosis, convolutional neural network, knowledge graph, attention mechanism

## Abstract

An effective fault diagnosis method of bearing is the key to predictive maintenance of modern industrial equipment. With the single use of equipment failure mechanism or operation of data, it is hard to resolve multiple complex variable working conditions, multiple types of fault and equipment malfunctions and failures related to knowledge and data. In order to solve these problems, a fault diagnosis method based on the fusion of deep learning with a knowledge graph is proposed in this paper. Firstly, the knowledge rules of bearing data is used for entity extraction. Next, the multiscale optimized convolutional neural network (MOCNN) proposed in this paper is used for fault classification to achieve relationship extraction. Finally, the fault diagnosis graph of the bearing is constructed for fault-assisted decision-making as well as the detailed display of fault information. According to experiment analysis, the fault diagnosis model based on MOCNN proposed in this paper, which integrates the end-to-end convolutional neural network and the attention mechanism, still achieves an accuracy of 97.86% under the data set of 160 types of faults. Compared with the deep learning models such as Resnet and Inception in the noise environment of multiple working conditions and variable working conditions, the model proposed in this paper not only shows a faster convergence speed and stable performance, but also a higher accuracy in evaluation indicators, which is beneficial to practical use.

## 1. Introduction

Rolling bearings are widely used in industrial production equipment, and their health status has a huge impact on the performance, stability and service life of the entire mechanical equipment [1,2]. Therefore, the fault diagnosis and condition assessment of rolling bearings are of great significance. Normally, the fault diagnosis methods of bearings are mainly divided into two types, one based on mechanism experience [3] and the other based on data driving [4]. Affected by both external and internal factors, the failure degradation of rolling bearings often presents nonlinearity and instability. It is difficult to achieve ideal fault diagnosis results by merely relying on failure mechanism knowledge or data-operating for state identification and fault diagnosis. Therefore, the bearing fault diagnosis method based on the fusion of a knowledge graph and deep learning indicates a new research direction.

The concept of a knowledge graph was first proposed by Google in 2012 to improve the search engine [5]. Compared with traditional knowledge bases, knowledge graphs represents richer semantic relationships, higher knowledge quality, and better visualization effects, providing great convenience for human-computer interaction. The knowledge graph is composed of entities, attributes and relationships. Knowledge graphs are widely used. For example, Oramas et al. [6] used knowledge graphs to recommend sounds and music; Li et al. [7] combined the knowledge graph with SQL databases, proposed the framework design of industrial software design and development process, and verified it in specific industrial scenarios; Liu et al. [8] used the integration method of deep learning and knowledge graph to prove the effectiveness of market prediction and made stock investments. However, due to the complexity of knowledge in the field of fault diagnosis, it is extremely difficult to construct and use knowledge graphs rationally. If we can construct the knowledge graph in an accurate way, it will play a key role in maintaining the health of mechanical equipment. Therefore, this paper will attempt to construct the knowledge graph of rolling bearing fault diagnosis in order to contribute to the health maintenance of equipment.

The construction of a bearing fault graph requires fusion with data. As a subset of machine learning, deep learning has made significant progress in computer vision [9], natural language processing [10], biology [11], and other fields, and it has also been widely used in the field of fault diagnosis, such as by Liang et al. [12], who proposed a gearbox fault diagnosis model that combines wavelet transform (WT) and CNN. It was verified on two gearbox datasets and achieved good results, but as the network got deeper, the model increased more, which caused performance degradation. Thus, it is difficult to achieve better results for more complex data. The proposal of the Inception network has changed people’s perception of relying on increasing the depth and width of the network to improve its performance. It changes the structure of the network so that the parameters reduce while the depth and width of the network increases [13], and the size of the convolution kernel will directly affect the performance of the network, and a single-sized convolution kernel can lead to incomplete extraction of information [14].

In response to the problems above, this paper proposes a fault diagnosis method based on the fusion of convolutional neural networks and knowledge graphs. The main contributions are as follows:(1)Extracting entities based on the equipment mechanism and knowledge rules of bearing data, then applying the MOCNN to categorize faults through data labeling to complete extraction of data relationship, and eventually, the knowledge graph of bearing faults is established. Furthermore, by forming the graph to assist decision-making as well as to display detailed fault information, it realizes a complete fault diagnosis, which moves away from a single reliance on mechanism knowledge or a data-driven diagnosis technique and toward the union of data and knowledge.(2)This paper offers an end-to-end one-dimensional multiscale convolutional neural network model (MOCNN) that combines advantages of one-dimensional convolutional networks in processing one-dimensional input. The model first implements sensitive feature extraction of the input 1D signals from different angles using the modified Inception module, and then adds a residual link to the Inception block to learn more abundant features. Following that, the retrieved features are fine-tuned using the channel attention and spatial attention modules. To improve the performance of the model and the effectiveness of defect diagnosis, the L2 regularization is introduced to the attention module and the classification layer.(3)The model is combined with the knowledge graph, marked by the CWRU data set, and adopted a new experimental division method. Compared with Resnet and Inception, this model has a good diagnosis impact. It is more suitable for large data and multi-classification problems and also features fast and stable convergence on small data sets and multi-classification problems. Moreover, it performs well in noise environment tests in various noise environments, working situations, and variable working conditions.

The rest of this paper is organized as follows: Related work is described in Section 2. Section 3 introduces the fault diagnosis method, fault diagnosis model and data processing proposed in this paper. In Section 4, entities and relationships are extracted and a knowledge graph is constructed. At the same time, the fault diagnosis model is compared and discussed from different angles. A conclusion is provided in Section 5.

## 2. Related Work

### 2.1. Knowledge Graph

The construction methods of the knowledge graph are mainly divided into bottom-up and bottom-down [15], and the bottom-up construction method is to perform entity extraction, relationship extraction and attribute extraction on the data first, and then perform knowledge fusion and entity alignment. That is, different representations of unified entities are denoised to obtain a unified representation, then entity disambiguation and attribute matching are performed, and finally ontology construction and knowledge inference are performed to form a knowledge graph. The bottom-down construction approach is to first create the top-level concept ontology and then extract entities and relationships from data sources to match and update them into the initially created top-level ontology. Since domain knowledge graphs require significant knowledge expertise, this paper uses a combination of both for construction.

### 2.2. CNN

A convolutional neural network is a multilayer feedforward neural network that updates the parameters of the network by a backpropagation algorithm and consists of convolutional layers, pooling layers, and fully connected layers [16]. The convolutional layer is mainly for extracting features automatically and the one-dimensional convolutional neural network has a strong ability to extract sequential features [17]; the pooling layer is utilized to further highlight the extracted features while reducing the dimensionality of the data, which can both speed up and prevent overfitting, and after multiple rounds of convolution and pooling, the extracted features are inputted to the fully connected layer to realize the classification task [18].

### 2.3. Multiscale Convolution

Rolling bearing vibration signals often produce non-periodic vibration signals with the appearance of damage points. A single-size convolution kernel lacks adaptability when extracting features and cannot be changed according to changes in input signals. Therefore, this paper uses a multiscale convolution layer to replace the single-scale convolutional layer in the conventional network, allowing the model to learn the nonlinear relationship between input and output through richer features, thereby improving the classification accuracy of each category. Figure 1 shows the common Inception network module.

### 2.4. Residual Learning

In response to the gradient disappearance and explosion problems in deep neural network training [19], He et al. first proposed residual neural networks [20] (Resnet) in 2016, designed residual blocks with shortcut connections, and used multiple residual blocks connected to build residual neural networks. The structure of residual blocks is shown in Figure 1, which is defined as follows:(1)$$H\left(x\right)=F\left(x\right)+x$$
where x and $H\left(x\right)$ represent the input and output functions respectively and F represents the residual function. The + in the figure is the summation of corresponding elements of the feature graph. The residual neural network is passed to the previous layer through a shortcut connection, thus effectively solving the problem of gradient disappearance along with the increasing depth during network training [21].

### 2.5. Attentional Mechanisms

CBAM is a combined spatial and channel attention mechanism network that learns along two independent dimensions of channel and space and obtains the corresponding weights [22]. Supposing that the input feature F is of size H×W×C, where H, W, C represent the length, width and number of channels of the feature, respectively. The channel attention is shown in Figure 2. The feature F is changed to 1×1×C after maximum pooling and average pooling, and then inputted to the multilayer perceptron (MLP) with shared weights. After that, by using the sigmoid activation function, the MLP processed and summed features are mapped and the weight coefficients $M_{C}\;$are obtained, which are then multiplied with the input features to get the new features after the channel attention.

The spatial attention is shown in Figure 3. The features F are spliced on the channel after maximum pooling and average pooling, respectively, obtaining the H×W×2 features, which are inputted to the convolution kernel of size 1 × 7 and the convolution layer with sigmoid activation function to obtain the weight coefficients $M_{s}$, which are then multiplied with the input features to obtain the new features after spatial attention.

### 2.6. Evaluation Metrics

In this study, accuracy, macro-precision, macro-recall and macro-$F_{1}$ score are used to comprehensively evaluate the classification performance of the model [23,24]. They are defined as follows:(2)$$Accuracy=\frac{TP+TN}{TP+FP+FN+TN}\times 100\%$$
(3)$$Precision=\frac{TP}{TP+FP}\times 100\%;\;Precision_{macro}=\frac{\sum_{i=1}^{L}Precision}{\left|L\right|}$$
(4)$$Recall=\frac{TP}{TP+FN}\times 100\%;\;Recall_{macro}=\frac{\sum_{i=1}^{L}Recall}{\left|L\right|}$$
(5)$$F1\;score_{macro}=\frac{2\times Precision_{macro}\times Recall_{macro}}{\;Precision_{macro}+Recall_{macro}}$$
where TP,FP,FN,TN represent true positive, false positive, false negative, and true negative, separately.

## 3. Proposed Method

This paper proposes a fault diagnosis method based on a knowledge graph and deep learning, which establishes a fault knowledge graph by extracting mechanistic knowledge and empirical rules of the equipment. The MOCNN model is capable of realizing data classification and can effectively combine mechanical knowledge with operating data of the equipment, and eventually forms a fault diagnosis model with the fusion of knowledge and data. Moreover, this model produces the knowledge graph to facilitate decision-making as well as the detailed display of fault information.

### 3.1. The Proposed Fault Diagnosis Model

The convolution layer is connected to the batch normalization (BN) layer after the convolution layer; the BN layer performs a batchwise mean reduction and divides the variance operation on the input data, which can speed up the training of the network [25] and increase the robustness of the model, as shown in Equations (6)–(9). After the BN operation on the dataset, the datasets are processed using the linear activation unit ReLU, and the definition of the ReLU activation function is shown in Equation (10): the batch normalization layer normalizes each batch of data to alleviate the problem of gradient disappearance in deep neural networks and improve the nonlinear expression of model pairs. The ReLU activation function can effectively improve the dispersion problem of the gradient and enhance the convergence of the network model efficiency [26].
(6)$$\mu =\frac{1}{N}\overset{N}{\underset{i=1}{\sum }}x_{i}$$
(7)$$\sigma^{2}=\frac{1}{N}\overset{N}{\underset{i=1}{\sum }}{\left(x_{i}-\mu_{i}\right)}^{2}$$
(8)$$k_{i}=\frac{x_{i}-\mu }{\sqrt{\sigma^{2}+\varepsilon }}$$
(9)$$y_{i}=\gamma k_{i}+\beta$$
where μ is the mean value of the samples, $x_{i}$ represents the characteristics of the inputs, N denotes the number of samples, σ is the variance, $k_{i}$ represents the normalized sample values, $y_{i}$ is the output of $k_{i}$ scaled, ε is a constant close to 0, and γ and β are scaling factors.
(10)$$Z_{t\;}^{l}=f\left(x_{t\;}^{l}\right)=max\left\{0,x_{t\;}^{l}\right\}$$
where $x_{t\;}^{l}$ represents the input and $Z_{t\;}^{l}$ represents the output.

In order to solve the problem of large computational complexity of larger convolutional kernels, this model first uses a 1 × 1 convolutional kernel to reduce the number of channels of the features and then performs the corresponding convolutional operations on them. 1 × 3 pooling can remove the redundant information extracted from the previous layer of convolution, and a larger convolutional kernel of 1 × 7 is set because the correlation of features may vary, and to learn this feature, a large convolutional kernel is required. The number of convolutional kernels in each branch of the module Basic Block A is 16. The improved Inception-Resnet module in this paper can not only overcome the gradient disappearance problem of deep network, but also solve the problem of optimal combination of convolution kernel size. At the same time, splitting the convolution kernel can increase the diversity of features and nonlinear expression. The number of convolutional kernels in the four branches of module Basic Block B are 16, 16, 16, 16, 32, and 32. The structures of modules Basic Block A and Basic Block B are shown in Figure 4.

Basic Block C combines the advantages of Resnet and CBAM to mitigate the loss and waste of information during network propagation as the network deepens and to focus attention on features that are more important for the signal classification task. This model incorporates L2 regularity for both spatial attention and channel attention mechanisms, and the structure of this module is illustrated in Figure 5.

By improving different modules, the structure of the MOCNN model constructed in this paper is shown in Figure 6.

The Figure 6 shows the structure of the MOCNN network proposed in this paper, which consists of two groups of convolutional layers, batch normalization layers and activation layers, among which the number of the first group of convolutional kernels is 32, the second group is 64 and both are of size 1 × 3. The number of convolutional kernels in the convolutional layer behind the Basic Block B module is 128, and its size is also 1 × 3. Basic Block C connects with the GlobalAveragePooling layer, then sets parameters of the Dropout layer to 0.5 to prevent the model from overfitting. In all convolutional layers, the step size is set to 1. Categorical Crossentropy is used as the loss function, Adam is chosen as the optimizer, the initial learning rate is set to 0.001, and the L2 regular term is added to the dense layer of the model to prevent overfitting.

### 3.2. The Proposed Fault Diagnosis Method

The fault diagnosis scheme proposed in this paper is mainly divided into three parts. The process is shown in Figure 7. First of all, the rules are extracted using the knowledge of equipment mechanism, and the rule nodes are matched with the node attributes to complete the entity extraction, then the rules are used as a classification basis and combined with the equipment operation data for labeling; the MOCNN model is trained through the training set and the test set is used for testing to complete the task of fault classification and to realize relation extraction. Finally, the triplet is constructed according to the entity and relationship, and the construction of the fault knowledge graph is completed through the relationship triplet.

### 3.3. Data Preparation

A large number of training samples can improve the generalization performance of the model, so this paper uses the data enhancement manner to expand the number of training samples. Taking the dataset in Section 4.1 as an example, the sliding segmentation method is used to achieve overlapping interception of bearing signals, with each training sample having a length of 1024 and a sliding interval of 238, resulting in a sample expansion of 500 samples in each category. The method maintains the continuity of the signal, and the principle is shown in Equation (11).
(11)$$N=\left[\frac{L-L_{0}}{S}+1\right]$$
where N is the number of samples, L is the length of the original vibration signal, $L_{0}$ is the length of each training sample, and S is the sliding interval of the sliding window.

## 4. Experimental Study and Analysis

This experiment includes the construction of KG and the classification experiments of fault data by MOCNN. The knowledge graph is constructed by using the Neo4j graph database for storage, and py2neo is used to implement the python program to operate Neo4j. Fault classification is performed using the MOCNN model to classify the data, and the performance of the model is also discussed.

### 4.1. Construction of Knowledge Graphs

#### 4.1.1. Entity Extraction

The dataset used for the experiments is the bearing dataset from Case Western Reserve University (CWRU), which is widely used in the field of fault diagnosis. The mechanistic knowledge of this dataset is mainly an introduction of the dataset, including motor load, motor speed, bearing fault location and fault size. The rules are summarized according to the dataset, which mainly include:

(1) The fault type is divided into the drive end with a sensor sampling frequency of 12 kHz and 48 kHz, the fan end with a sampling frequency of 12 kHz, and the normal type of data; (2) The fault size of the drive end and the fan end is divided into 0.007 inches, 0.014 inches, 0.021 inches, and some are 0.028 inches; (3) The faults on the drive side and fan side were generated at 0 HP,1 HP,2 HP, and3 HP motor loads, and the normal operation data were also collected under four loads corresponding to rotational speeds of 1797 r/min, 1772 r/min, 1750 r/min, 1730 r/min, respectively; (4) Bearing failures at the drive end and fan end include rolling element, inner ring and outer ring failures, respectively, and the outer ring includes failures at 3, 6 and 12 o’clock. Symbolizing the name according to the rules above, all together there are six levels of fault nodes: the zero fault node can be set as a test bed as the central node of the knowledge graph; the first level fault node (Level_1): Type, a total of 4 categories; the second level fault node (Level_2): Size, 10 types in total; Level 3 failure node (Level_3): Load, 44 types in total; Level 4 failure node (Level_4): Location, 119 types in total; Level 5 failure node (Level_5): Clock, 160 in total. The attributes of the nodes entered into the knowledge graph include Node ID, Node Label, Node Type, and Node Description. Some fault nodes are shown in Table 1:

#### 4.1.2. Relation Extraction

The MOCNN model constructed in this paper is used to classify the fault data, and the data set is divided into four classes, 10 classes, 44 classes, 119 classes and 160 classes, respectively, according to the type of classification nodes for data labeling. 1024 data points are taken as a time series segment, and a sliding window is used for data enhancement, and then the training set and test set are randomly divided in the ratio of 7:3 and the results of model testing are shown in Figure 8 and Table 2:

As shown in Table 3, DE12 has the largest amount of data (30,000, 1024), while NO has the least amount of data (2000, 1024), which represents a big difference in the amount of data. However, it converges quickly and achieves 100% accuracy, which proves the advantage of the model. Next is the second level fault node classification, and no normal data is added to this level of classification as normal data does not have fault characteristics. As shown in Table 4, there are 10 categories of data. Although the amount of data is unbalanced, the model quickly reached 99% accuracy, with slight fluctuations compared to the level 1 fault classification curve; the third level fault classification curve converges more slowly compared to the second one, the accuracy reaches 97.63%; the accuracy of the fourth level fault classification shows a slight decrease due to a sharp increase in classification categories in the face of unbalanced data volumes. Nevertheless, the accuracy still reaches 93.35%, which is a very impressive result in practical engineering applications; the data volume of the fifth level fault classification is relatively more balanced since each category is (500, 1024), the accuracy reaches 97.86% and the other evaluation indexes are basically the same, achieving a better fault diagnosis result.

Confusion matrix plot of Level_1(Type) and Level_2(Size) classification results are shown in Figure 9.

The GRU network is a model that maintains the LSTM effect with a simpler structure, fewer parameters and better convergence performance. The algorithm proposed in this paper is compared with different models in Level_5(Clock) in the fifth-level classification fault experiment. The results are shown in Table 5.

From the results, it can be obtained that the model has a great improvement in various evaluation metrics compared to the traditional machine learning algorithms. Although the DRNN, GRU and Inception network also show great improvement, they still have a large gap compared with the model proposed in this paper.

It can be seen that the MOCNN model has the comprehensive advantages of high accuracy, fast convergence speed and stable classification in multi-classification problems and achieves a better fault diagnosis effect. To analyze in a specific way, the beginning convolution layer, BN layer and Basic Block A play the role of feature extraction, thereby improving the convergence speed; Basic Block B avoids gradient disappearance while extracting features, making the model more stable; Basic Block C performs feature extraction by assigning weights, thus improving the training speed and convergence speed of the model. The final Dropout layer and L2 regularity effectively mitigate the overfitting problem and improve the generalization ability of the model.

#### 4.1.3. Graph Construction

The extracted nodes and relationships can be expressed through triples as shown in Table 6. The entity nodes and relationship nodes are imported into the Neo4j database using a Python program and the knowledge graph constructed can be manipulated utilizing py2neo and Cypher statements. The graph can show details of fault nodes and other nodes that are related to them can be clearly seen.

As shown in Figure 10, neo4j can set the number of levels and nodes displayed. The figure shows a total of 30 nodes with a total of four levels. The red point in the center is Level_0: Equipment, the four yellow points are Level_1, which are DE12, DE48, FE12, Normal, the orange is Level_2, and the blue is Level_3 and the connections at each level can be clearly observed. The Figure 11 shows some nodes intercepted when all nodes are displayed. When a node is selected, such as the outermost green node, the information of the node, such as id, label and name, will be displayed on the interface. At the same time, the Cypher language can be used to query the nodes.

For example, to query all nodes connected to the DE12_size7_load0 node, we can enter the command in neo4j.

Match p=(n:level_3{name: “DE12_size7_load0”})-:rel{relation: “belongs to”}]->() return p; then all the nodes connected to that node will be displayed. The graph can also perform a knowledge quiz to return information about faults to aid decision-making and improve the efficiency of fault diagnosis.

### 4.2. Model Performance Discussion

The model MOCNN achieves better fault diagnosis results with a large amount of data. This paper also tests the model in the conventional fault diagnosis experimental method, mainly for exploring the dataset under DE12 working conditions, and the data are processed and divided in the same way as described above.

#### 4.2.1. Performance of MOCNN on Small Datasets

First, the data sets under 0 HP(1797), 1 HP(1772), 2 HP(1750), and 3 HP(1730) conditions are enhanced and divided to obtain the datasets Dataset A, Dataset B, Dataset C, and Dataset D, respectively. The four datasets are combined to obtain Dataset E, as shown in Table 7, and then the model is used to conduct 10 types of experiments. The results are shown in Table 8.

From Table 8, it can be seen that the accuracy rate of this model on the four small datasets has reached 100%, the diagnostic effect of 99.98% is also achieved on dataset E and the accuracy rate and loss curve can be obtained. The model converges quickly and then stabilizes after 50 epochs after a small fluctuation from the Figure 12.

Using T-SNE to visualize the changes in the classification results of MOCNN in different layers (under 3 HP conditions) as shown in Figure 13, the data in Figure 13a passes through the Input layer, and the cluttered distribution of the data can be observed. After passing through Basic Block A and Basic Block B, samples of the same type slowly converge together, and after passing through Basic Block C, a good classification effect has been achieved.

#### 4.2.2. Anti-Noise Performance of MOCNN

Since the rolling bearing working environment is often accompanied by noise variations, a variable noise working environment is set up to verify that the method proposed in this paper has noise immunity. The definition formula of signal-to-noise ratio is shown in Equation (12). Next, we add different levels of noise to the above datasets A, B, C, and D for testing. It can be seen that the model has achieved 100% accuracy at 2 dB, 0 dB, and −2 dB, indicating that the model has good filtering and anti-noise functions. Meanwhile, the model also achieved 98.2% of the fault diagnosis results under the operating condition of −10 dB. The results are shown in Table 9 and Figure 14.
(12)$$SNR_{db}=10\log_{10}\frac{P_{signal}}{P_{noise}}$$

$P_{signal}$ and $P_{noise}$ are the power of the signal and the power of the noise, respectively.

#### 4.2.3. Comparison with Other Algorithms

In order to fully illustrate the advantages of the model’s noise immunity function, it is compared with other algorithms adding different noises in Dataset D and Dataset E, adding −10 dB noise, and the results are shown in Table 10. It can be seen that the model has the highest fault diagnosis accuracy in all four noise environments and the best test results in Dataset E, which fully illustrates the superiority of the model’s noise immunity performance.

In order to fully illustrate the advantages of the model’s anti-noise function, the results of adding different noise to Dataset D compared with other algorithms are shown in Table 10 and Figure 15. Adding −10 dB noise to Dataset E for comparison, and the results are shown in Figure 16. It can be seen that the model has the highest fault diagnosis accuracy in the four noise environments, and the test results in Dataset E are also the best. Fully explain the superiority of the model’s anti-noise performance

#### 4.2.4. Performance of the Model under Variable Load Conditions

With regard to the analysis of fault diagnosis results under variable load conditions, Datasets A, B and C, were trained at −10 dB signal-to-noise ratio, and then the fault diagnosis capability of the model under different conditions was verified by using test set data under three other load conditions. The results are shown in Figure 17.

The complexity of the model is an important index to measure the bearing fault diagnosis method. The time complexity determines the testing time of the model, and the space complexity determines the number of parameters of the model. Table 11 summarizes the number of parameters and testing time of each model, from which we can notice that MOCNN has a very light structure, uses a small number of parameters with better performance than Resnet, and has basically the same testing time as the Inception10 network, proving that MOCNN has a high parameter utilization rate, so this model is well suited for application in engineering practices.

## 5. Conclusions

In this paper, we propose a deep learning and knowledge graph fusion method for bearing fault diagnosis which combines the mechanistic knowledge of rolling bearings with vibration signal data to improve the efficiency of fault diagnosis. By using this method, the bearing data is firstly extracted by rules, entities are extracted, then the data is labeled and the optimized convolutional neural network model MOCNN is used to classify faults, realize relation extraction and obtain triad for knowledge graph construction to assist fault diagnosis decision and visualize the results. In addition, this paper also compares and discusses the models. Using the CWRU dataset, a new classification method can still achieve an accuracy of 97.86% when the faults are up to 160 classes, and it is fully compared with other models in environments such as multiple operating conditions and noise environments. The experimental results show that the one-dimensional fault diagnosis model proposed in this paper, regardless of the size of the dataset, not only has a fast convergence rate and stable operation, but also has the best noise immunity and migration performance, which proves the practicality of the fault diagnosis model proposed.

This study provides a new solution for fault diagnosis of mechanical equipment and also illustrates the difficulty of diagnosing compound faults in industrial equipment. The model in this paper is currently only for historical data, and in the future, we will also try real-time diagnosis and add other kinds of data to build a richer knowledge graph using deep learning algorithms.

## Figures and Tables

**Figure 1 entropy-24-01589-f001:**
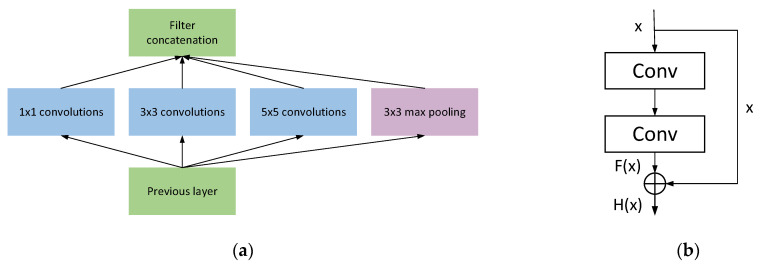
The common Inception network module and the structure of residual blocks: (**a**) Diagram of the Inception structure; (**b**) Diagram of the residual structure.

**Figure 2 entropy-24-01589-f002:**
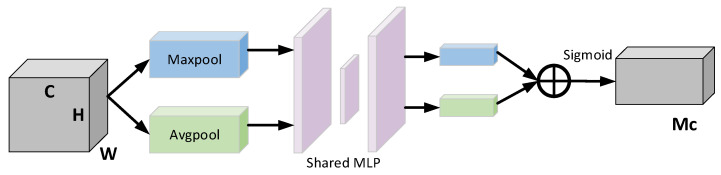
Channel Attention Module.

**Figure 3 entropy-24-01589-f003:**
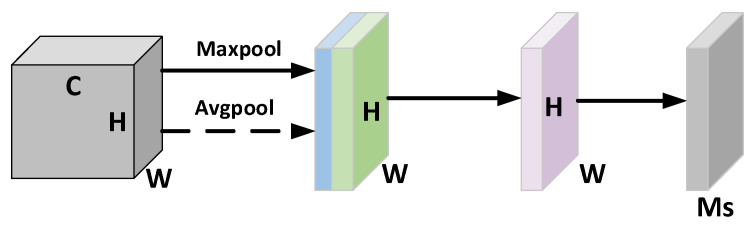
Spatial Attention Module.

**Figure 4 entropy-24-01589-f004:**
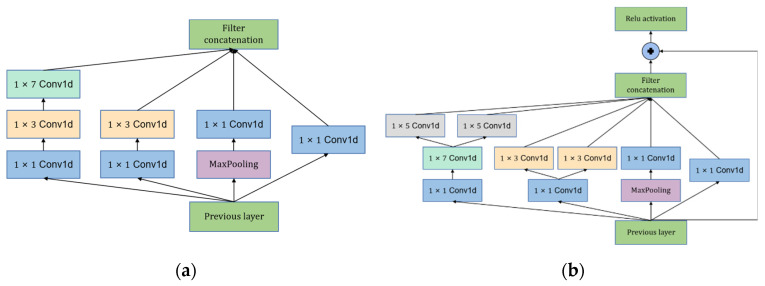
The structures of modules Basic Block A and Basic Block B: (**a**) Basic Block A; (**b**) Basic Block B.

**Figure 5 entropy-24-01589-f005:**
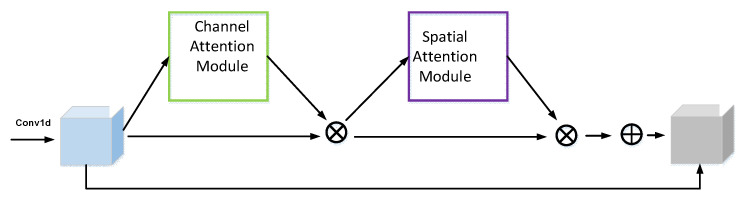
Basic Block C.

**Figure 6 entropy-24-01589-f006:**
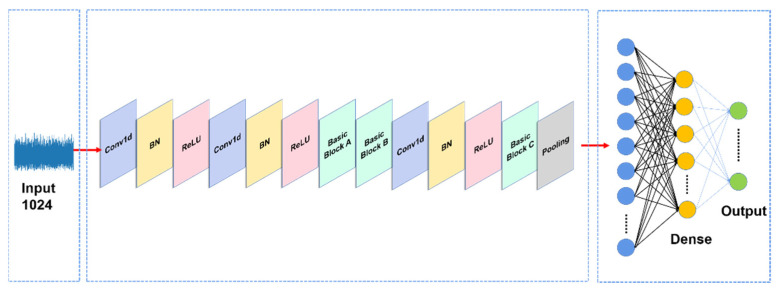
The diagram of the proposed MOCNN structure.

**Figure 7 entropy-24-01589-f007:**
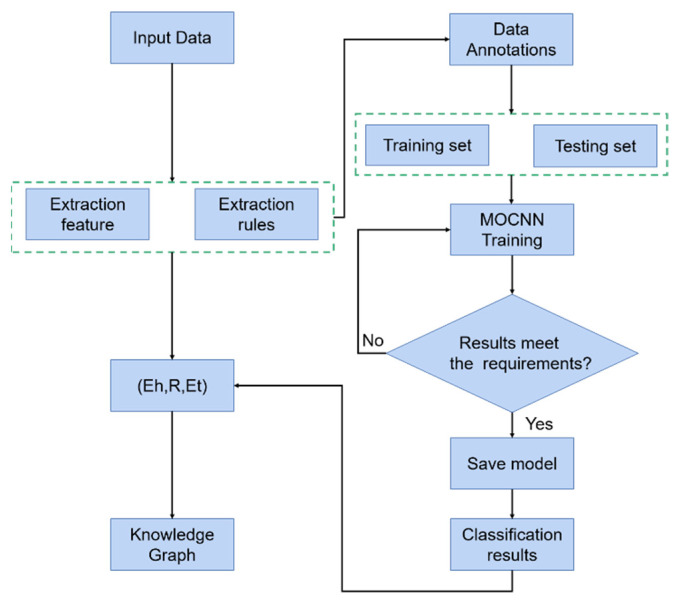
The process diagram of the proposed method for the bearing fault diagnosis.

**Figure 8 entropy-24-01589-f008:**
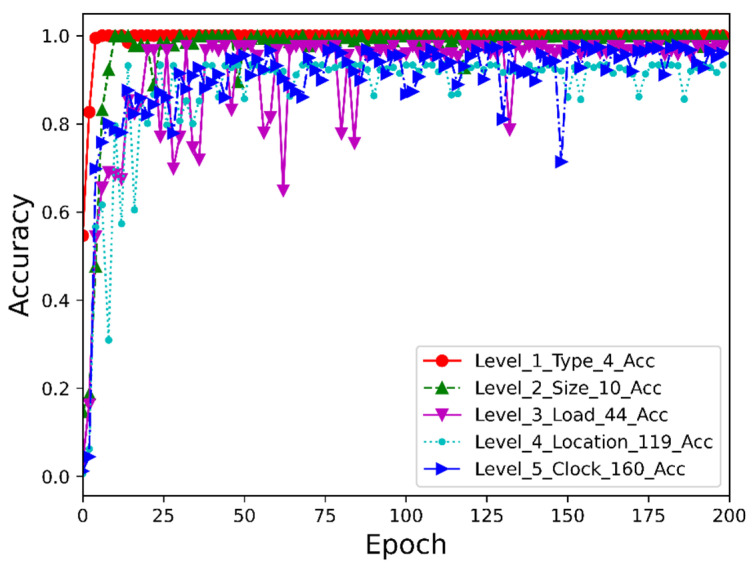
Fault classification results at different levels of MOCNN.

**Figure 9 entropy-24-01589-f009:**
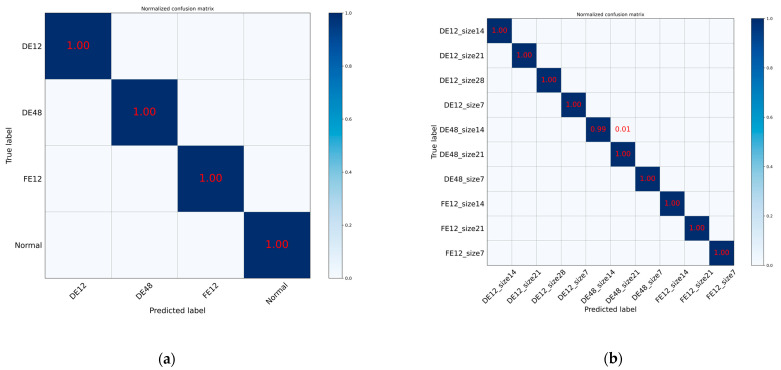
The confusion matrix of the MOCNN model for the Level_1 and Level_2 samples: (**a**) Level_1(Type); (**b**) Level_2(Size).

**Figure 10 entropy-24-01589-f010:**
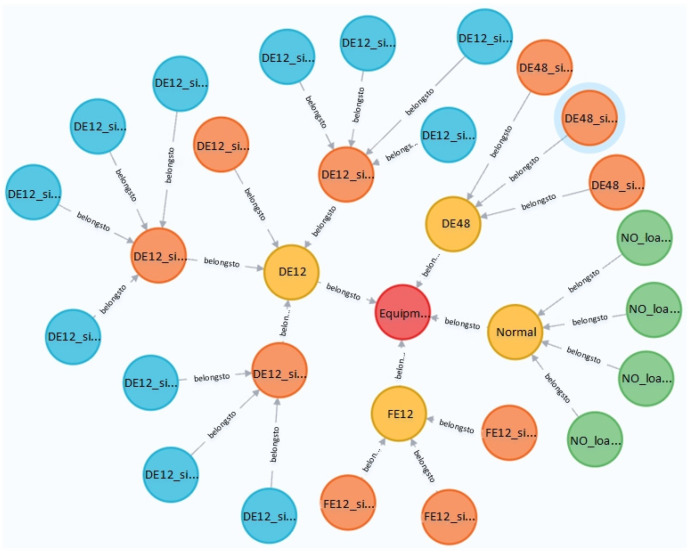
Results of the fault type.

**Figure 11 entropy-24-01589-f011:**
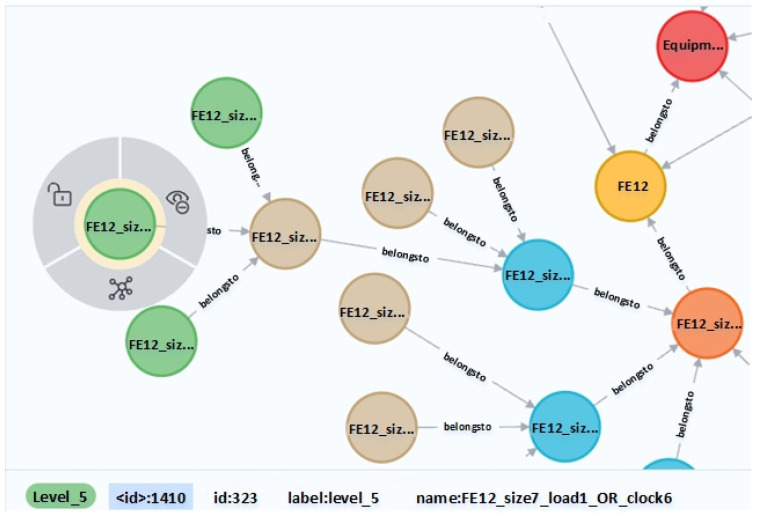
Node information of knowledge graph.

**Figure 12 entropy-24-01589-f012:**
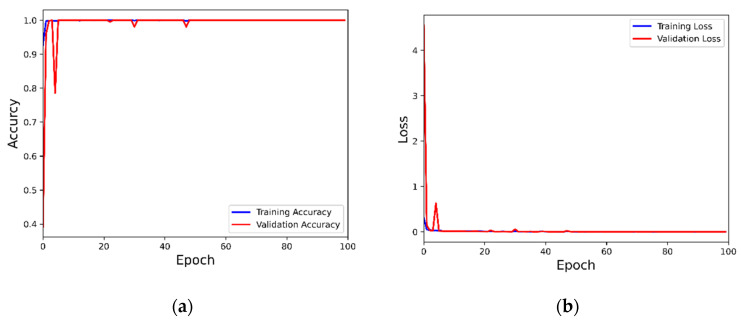
Training accuracy and loss of the MOCNN model for the dataset E: (**a**) The classification accuracy of MOCNN on dataset E; (**b**) The loss of MOCNN on dataset E.

**Figure 13 entropy-24-01589-f013:**
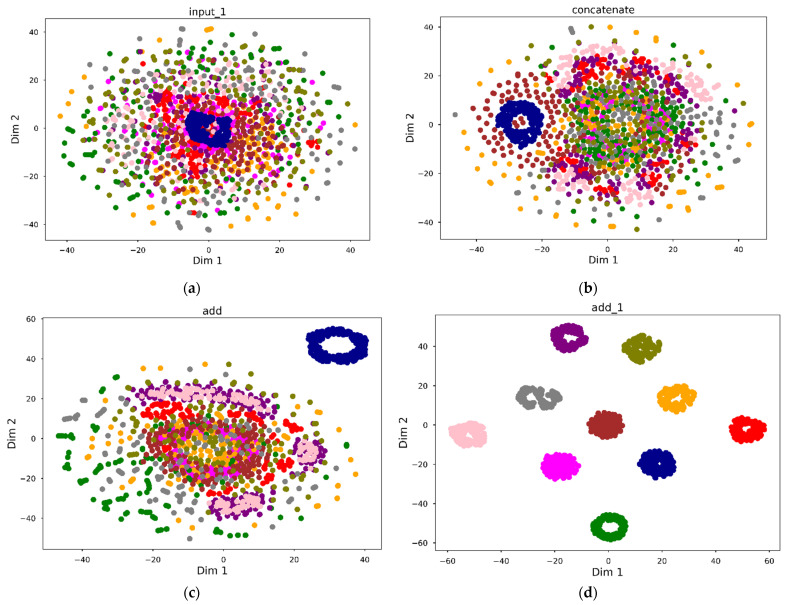
3 HP two-dimension distribution images.

**Figure 14 entropy-24-01589-f014:**
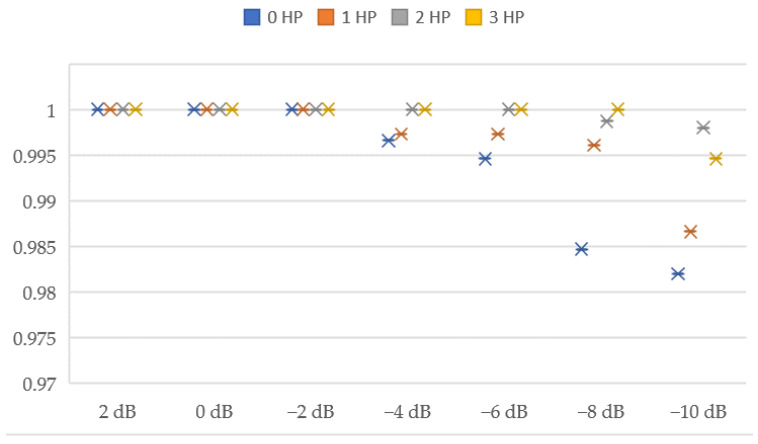
Test results of MOCNN model under different working conditions and different noise environments.

**Figure 15 entropy-24-01589-f015:**
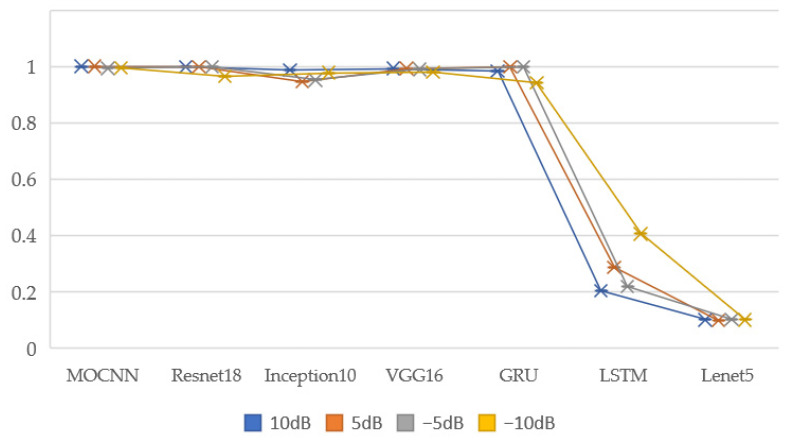
Test results of different algorithms in different noise environments.

**Figure 16 entropy-24-01589-f016:**
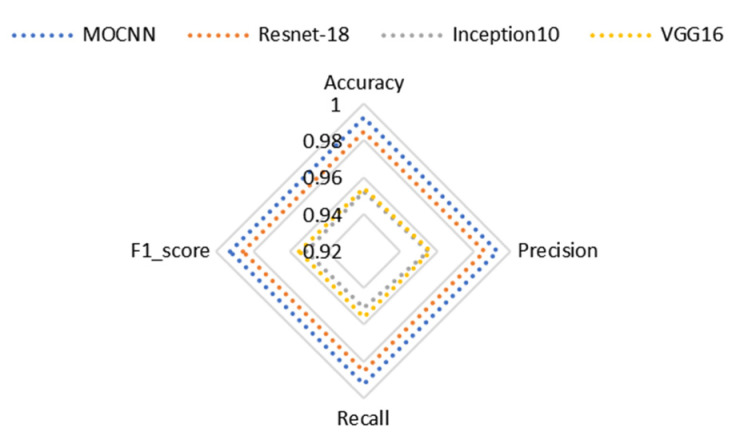
Test results of different algorithms under −10 dB dataset E.

**Figure 17 entropy-24-01589-f017:**
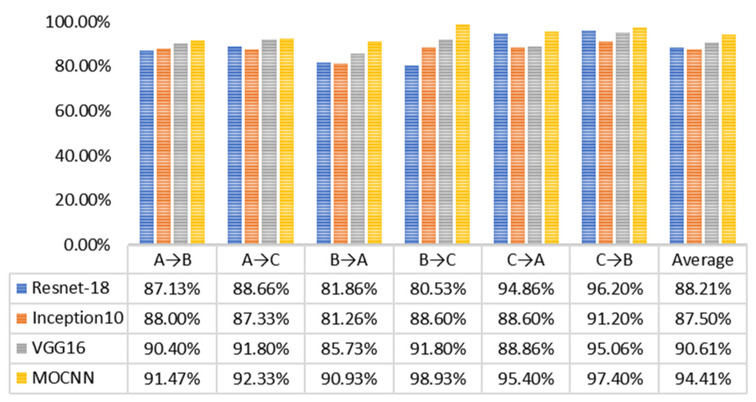
Comparison of results of different algorithms in off design experiments.

**Table 1 entropy-24-01589-t001:** Some fault nodes of knowledge graph.

Node ID	Node Type	Node Label	Node Description
0	Level_0	Equipment	Experimental Equipment
1	Level_1	DE12	12DriveEndFault
8	Level_2	DE12_size7	12DriveEndFault_FaultDiameter0.007
27	Level_3	DE12_size7_load0	12DriveEndFault_FaultDiameter0.007_Load0HP
93	Level_4	DE12_size7_load0_OR	12DriveEndFault_FaultDiameter0.007_Load0HP_OuterRace
222	Level_5	DE12_size7_load0_OR_clock6	12DriveEndFault_FaultDiameter0.007_Load0HP_OuterRace_@6:00

**Table 2 entropy-24-01589-t002:** Fault classification results at different levels of MOCNN.

	Accuracy (%)	Macro-Precision (%)	$\mathrm{Macro}-F_{1}\;\mathrm{Score}\;(\%)$	Macro-Recall (%)	Fault Type
Level_1(Type)	100	100	100	100	4
Level_2(Size)	99.92	99.93	99.23	99.23	10
Level_3(Load)	97.63	97.78	97.64	97.65	44
Level_4(Location)	93.35	93.35	93.35	93.35	119
Level_5(Clock)	97.86	97.86	97.86	97.86	160

**Table 3 entropy-24-01589-t003:** The number of Level_1(Type) samples.

Type	DE12	DE48	FE12	NO
Number	30,000	25,500	22,500	2000

**Table 4 entropy-24-01589-t004:** The number of Level_2(Size) samples.

Type	DE12_Size7	DE12_Size14	DE12_Size21	DE12_Size28	DE48_Size7	DE48_Size14	DE48_Size21	FE12_Size7	FE12_Size14	FE12_Size21
Number	10,000	6000	10,000	4000	10,000	5500	10,000	10,000	6500	6000

**Table 5 entropy-24-01589-t005:** Results of the different algorithm models for the Level_5 samples.

	Metric	Accuracy (%)	Macro-Precision (%)	$\mathrm{Macro}-F_{1}\;\mathrm{Score}\;(\%)$	Macro-Recall (%)
Method					
XGBoost		19.31	18.63	17.37	19.28
RF		54.20	54.66	53.23	54.38
DRNN		70.26	71.47	68.41	70.27
GRU		83.58	83.77	83.65	82.74
Inception10		86.23	86.92	86.27	85.67
MOCNN		97.86	97.86	97.86	97.86

**Table 6 entropy-24-01589-t006:** Partial fault relation nodes.

Start Entity	Tail Entity	Relation
DE12	Equipment	belongs to
DE12_size7	DE12	belongs to
DE12_size7_load0	DE12_size7	belongs to
DE12_size7_load0_OR	DE12_size7_load0	belongs to
DE12_size7_load0_OR_clock6	DE12_size7_load0_OR	belongs to

**Table 7 entropy-24-01589-t007:** Division of CWRU dataset.

Fault Type	Class Label	Dataset A Training/Test	Dataset B Training/Test	Dataset C Training/Test	Dataset D Training/Test	Dataset E Training/Test
DE12_size7_load0_BA	B0.007	3500/1500	3500/1500	3500/1500	3500/1500	14,000/6000
DE12_size14_load0_BA	B0.014	3500/1500	3500/1500	3500/1500	3500/1500	14,000/6000
DE12_size21_load0_BA	B0.021	3500/1500	3500/1500	3500/1500	3500/1500	14,000/6000
DE12_size7_load0_IR	IR0.007	3500/1500	3500/1500	3500/1500	3500/1500	14,000/6000
DE12_size14_load0_IR	IR0.014	3500/1500	3500/1500	3500/1500	3500/1500	14,000/6000
DE12_size21_load0_IR	IR0.021	3500/1500	3500/1500	3500/1500	3500/1500	14,000/6000
DE12_size7_load0_OR	OR0.007	3500/1500	3500/1500	3500/1500	3500/1500	14,000/6000
DE12_size14_load0_OR	OR0.009	3500/1500	3500/1500	3500/1500	3500/1500	14,000/6000
DE12_size21_load0_OR	OR0.021	3500/1500	3500/1500	3500/1500	3500/1500	14,000/6000
NO_load0	NO	3500/1500	3500/1500	3500/1500	3500/1500	14,000/6000

**Table 8 entropy-24-01589-t008:** Test results of model MOCNN on the above five datasets.

	Accuracy (%)	Macro-Precision (%)	$\mathrm{Macro}-F_{1}\;\mathrm{Score}\;(\%)$	Macro-Recall (%)
3 HP	100	100	100	100
2 HP	100	100	100	100
1 HP	100	100	100	100
0 HP	100	100	100	100
Mix Load	99.98	99.98	99.98	99.98

**Table 9 entropy-24-01589-t009:** Test results of MOCNN model in different working conditions and different noise environments.

Load		SNR	2 dB	0 dB	−2 dB	−4 dB	−6 dB	−8 dB	−10 dB
	Metric								
0 HP	Accuracy (%)		100	100	100	99.66	99.46	98.47	98.2
	Macro-precision (%)		100	100	100	99.67	99.48	98.49	98.2
	Macro-recall (%)		100	100	100	99.66	99.46	98.49	98.2
	Macro-$F_{1}$ score (%)		100	100	100	99.67	99.47	98.48	98.2
1 HP	Accuracy (%)		100	100	100	99.73	99.73	99.61	98.66
	Macro-precision (%)		100	100	100	99.73	99.74	99.61	98.68
	Macro-recall (%)		100	100	100	99.73	99.73	99.60	98.68
	Macro-$F_{1}$ score (%)		100	100	100	99.73	99.73	99.60	98.68
2 HP	Accuracy (%)		100	100	100	100	100	99.87	99.8
	Macro-precision (%)		100	100	100	100	100	99.87	99.8
	Macro-recall (%)		100	100	100	100	100	99.87	99.8
	Macro-$F_{1}$ score (%)		100	100	100	100	100	99.87	99.8
3 HP	Accuracy (%)		100	100	100	100	100	100	99.46
	Macro-precision (%)		100	100	100	100	100	100	99.46
	Macro-recall (%)		100	100	100	100	100	100	99.46
	Macro-$F_{1}$ score (%)		100	100	100	100	100	100	99.46

**Table 10 entropy-24-01589-t010:** Test results of different algorithms in different noise environments.

SNR	Metric	MOCNN	Resnet-18	Inception10	VGG16	Lenet5	GRU	LSTM
10 dB	Accuracy (%)	100	99.89	98.66	99.26	10.05	98.33	20.33
	Macro-precision (%)	100	99.89	98.66	99.30	0.98	98.31	15.13
	Macro-recall (%)	100	99.89	98.60	99.26	10.05	98.30	20.06
	Macro-$F_{1}$ score (%)	100	99.89	98.62	99.27	1.79	98.28	8.06
5 dB	Accuracy (%)	100	99.89	94.67	99.26	9.8	99.93	28.66
	Macro-precision (%)	100	99.89	95.32	99.26	0.98	99.93	27.25
	Macro-recall (%)	100	99.89	94.53	99.26	10.01	99.93	28.48
	Macro-$F_{1}$ score (%)	100	99.89	94.38	99.26	1.78	99.93	0.1983
−5 dB	Accuracy (%)	99.93	99.93	95.2	98.93	10.13	99.93	21.8
	Macro-precision (%)	99.93	99.93	95.59	98.95	10.13	99.93	13.83
	Macro−recall (%)	99.93	99.92	95.15	98.93	10.03	99.93	21.72
	Macro-$F_{1}$ score (%)	99.93	99.92	94.83	98.92	1.84	99.93	13.57
−10 dB	Accuracy (%)	99.46	96.43	97.73	98.01	10.03	94.13	40.53
	Macro-precision (%)	99.46	96.42	97.79	98.02	1.02	95.29	37.49
	Macro-recall (%)	99.46	95.88	97.69	98.01	10.03	93.99	39.98
	Macro-$F_{1}$ score (%)	99.46	95.83	97.72	97.98	1.81	94.18	30.71

**Table 11 entropy-24-01589-t011:** Number of parameters and test time for the different algorithms.

	MOCNN	Resnet-18	Inception10	VGG16
Number of Parameter	$2.6\times 10^{5}$	$3.9\times 10^{6}$	$5.4\times 10^{4}$	$1.3\times 10^{7}$
Test Time/s	0.613	2.598	0.504	0.533

## Data Availability

The experimental data are available from the Bearing Data Center of Case Western Reserve University deposited in https://engineering.case.edu/bearingdatacenter/welcome (accessed on 1 May 2022).

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
