# Peer review of "Bearing Fault Diagnosis Method Based on Convolutional Neural Network and Knowledge Graph"

_entropy, 2022, doi:10.3390/e24111589_

Round 1

Reviewer 1 Report

The paper presents merit, and the results contribute to the relevant literature. I have the following remarks:

- Please streamline the Introduction: the exposition should be polished and the extent should be considerably decreased.

- Importantly, the contributions of this work should be mentioned in a numbered manner inside the text in the Introduction, so that the reader could relate to the new novel results. Under this form, the Introduction is too long and the reader never gets a grip of the real contribution of this paper.

- As a model robustness checking approach, I want to see an extensive empirical investigation - aside from the MOCNN and Resnet - a comparative evaluation vis-a-vis other established Machine Learning techniques, e.g., Type-3 Neurofuzzy models, Deep Recurrent Neural Networks etc. with the fusion of Knowledge Graphs.

- In the Conclusion section, please mention future applications and the future path of your method. I would like to see how it could be utilized in other/alternative or substitute theoretical or empirical studies.

- More importantly, the paper requires an extensive proofreading. The use of the English language is not suitable for such prestigious Journal. Please proceed to proofreading by a native English speaker.

Reviewer 2 Report

the paper is interesting.
My main concern is about the application in the industry, the methodology shown in figure 6 it is complex. In what kind of devices can this methodology be implemented?
a real-time or online detection can be reached?
the methodology can be implemented for a different signal, for example, current or sound, maybe temperature?

by obtaining rating values of 100%, the network is not over adjusted?

how to justify the use of a complex methodology given that there are works in the state of the art that obtain good results with simpler methodologies.?

Round 2

Reviewer 2 Report

Thank you for the answers